# The Application of Transition Metal Sulfide Nanomaterials and Their Composite Nanomaterials in the Electrocatalytic Reduction of CO$_2$: A Review

Jason Parsons [1],* and Mataz Alotaibi [2]

[1] Department of Chemistry, University of Texas Rio Grande Valley, 1 W University Blvd., Brownsville, TX 78521, USA
[2] Department of Mechanical Engineering, University of Texas Rio Grande Valley, 1201 W University Dr., Edinburg, TX 78539, USA
* Correspondence: jason.parsons@utrgv.edu; Tel.: +1-(956)-882-7772

**Abstract:** Electrocatalysis has become an important topic in various areas of research, including chemical catalysis, environmental research, and chemical engineering. There have been a multitude of different catalysts used in the electrocatalytic reduction of CO$_2$, which include large classes of materials such as transition metal oxide nanoparticles (TMO), transition metal nanoparticles (TMNp), carbon-based nanomaterials, and transition metal sulfides (TMS), as well as porphyrins and phthalocyanine molecules. This review is focused on the CO$_2$ reduction reaction (CO$_2$RR) and the main products produced using TMS nanomaterials. The main reaction products of the CO$_2$RR include carbon monoxide (CO), formate/formic acid (HCOO$^-$/HCOOH), methanol (CH$_3$OH), ethanol (CH$_3$CH$_2$OH), methane (CH$_4$), and ethene (C$_2$H$_4$). The products of the CO$_2$RR have been linked to the type of transition metal–sulfide catalyst used in the reaction. The TMS has been shown to control the intermediate products and thus the reaction pathway. Both experimental and computational methods have been utilized to determine the CO$_2$ binding and chemically reduced intermediates, which drive the reaction pathways for the CO$_2$RR and are discussed in this review.

**Keywords:** electrocatalysis; electrocatalytic CO$_2$ reduction reaction; transition metal sulfide; nanomaterials; composite nanomaterials



## 1. Introduction

Electrocatalysis can be defined as the use of electricity to perform chemical reactions at an electrode surface where the catalyst reduces the overall potential of the reaction, or the catalyst lowers the energy of the transition state for the reaction. In recent years, electrocatalysis has gained much interest in different fields such as catalysis [1], photocatalysis [1], energy storage [2,3], environmental remediation [4–8], material synthesis [9–11], gas sensing [12], and electrochemistry [13]. The application of nanomaterials in the aera of electrocatalysis has become more significant due to higher reactivity rates than that of the respective bulk materials [14,15]. The increased reaction rates are typically due to the high surface area to volume ratio of nanomaterials. Areas such as photocatalytic hydrogen evolution have been advanced in recent years due to the use of composite nanomaterials [16]. The mechanisms or reactions and predictability of catalysts whether used in traditional or electrocatalytic systems remain the same. The Sabatier principle for catalytic activity and the choice of a desired catalyst are still applicable in electrocatalysis. The optimal catalytic activity occurs when the interactions of the catalyst and reaction substrate are of intermediate strength [17–19].

The most extensively studied reactions in the last two decades were the hydrogen evolution reaction (HER), hydrogen oxidation reaction (HOR), the oxygen evolution reaction (OER), the water splitting reaction, oxygen reduction reaction (ORR), CO$_2$ reduction

reaction ($CO_2RR$), methanol oxidation reaction (MOR), and nitrate reduction/ammonia production reactions [20–25]. These reactions are essential for fuel production, electricity generation in fuel cells, as well as environmental remediation. These reactions are similar to those observed in photocatalysis, where reactive oxygen species are generated at a catalytic surface which in turn oxidize the organic pollutant. In aqueous-based systems, the HER and OER are extremely important for environmental remediation.

The reaction in electrocatalysis can be initially based on an adsorption mechanism [26,27]. The adsorption is controlled by the attraction forces generated typically by an electrostatic interaction. The adsorption mechanism is followed by either oxidation or a reduction reaction (electron transfer reactions) followed by the desorption of the product molecule from the surface [28–30]. These steps are interdependent upon each other and sometimes the release of the bound molecule from the surface is slow and can inhibit the reaction progression. Furthermore, electrocatalytic reactions occur at the interface between the charged electrode surface and the electrolyte supplying a counter charge in solution. The surface charge is dependent on the electrode material, the potential applied to the electrode, the electrolyte, and the entire cell [31].

The electrocatalytic reduction mechanism of $CO_2$ involves multiple electron transfer steps as well as adsorption processes [32]. There has been much interest in the conversion of $CO_2$ into value-added products, which include formaldehyde, formic acid, and methanol, among other products [33–35]. Research has led to the production of ethanol, ethylene, methane, and syngas [34]. These types of products have been achieved using different catalysts; for example, $InO_x$, β-$Bi_2O_3$, Cu nanoparticles (NPs), Ag/Cu (NPs), Cu-Ce(OH)$_x$, Cu/nanodiamonds, CuO, Pd/B/C, Pd nanosheets, Pd/Mg nanospheres, Pd/Ag alloys, and AuPd-MOFs [35–47]. Oher examples of nanomaterials used in $CO_2$ reduction include monometallic nanoparticles, Au, Ag, Zn, Fe, Ni, Pt, Sn, In, and Pb [48–56]. Bimetallic nanomaterials have been also shown to be effective in the reduction of $CO_2$, including AuCu, PdPt, and CuI [57–59], while non-metallic catalysts including $MoS_2$, C-nanofibers, $C_3N_4$, N-doped nanodiamond, and B-doped diamonds [60] have been also used as catalysts for $CO_2$ reduction.

The most common reactions in the $CO_2$ reduction are shown below [60–62]:

$$CO_2 + 2H^+ + 2e^- \rightarrow CO + H_2O \quad E^0 = -0.53 \text{ V} \tag{1}$$

$$CO_2 + 2H^+ + 2e^- \rightarrow HCOOH \quad E^0 = -0.61 \text{ V} \tag{2}$$

$$CO_2 + 4H^+ + 4e^- \rightarrow HCOH + H_2O \quad E^0 = -0.48 \text{ V} \tag{3}$$

$$CO_2 + 6H^+ + 6e^- \rightarrow CH_3OH + H_2O \quad E^0 = -0.38 \text{ V} \tag{4}$$

$$CO_2 + 8H^+ \, 8e^- \rightarrow CH_4 + 2H_2O \quad E^0 = -0.24 \text{ V} \tag{5}$$

$$2CO_2 + 12H^+ + 12e^- \rightarrow C_2H_4 + 4H_2O \quad E^0 = -0.34 \text{ V} \tag{6}$$

These reactions are typically performed in a series of steps, with single electron transfers occurring. Typically, the reaction starts with the binding of the $CO_2$ molecule and the formation of radical species, which control the reaction pathway. For example, via the formation of the HCOO$^*$ radical, which forces the reaction to produce formate/formic acid. However, the formation of COOH$^*$ radical species typically forces the reaction to go through the pathway for the formation of CO as the final product [63]. The formation of methanol requires the formation of the CO$^*$ radical species followed by the formation of the CHO$^*$ and the HCHO$^*$ radicals. Subsequently, further hydrogenation can occur to form methanol ($CH_3OH$) [64]. Ethanol synthesis from $CO_2$ requires the hydrogenation of $CO_2$ to $CH_3^*$ radicals (which may come from methanol degradation) followed by the coupling of a $CO_2^*$ with the $CH_3^*$, hence forming the $CH_3COO^*$ intermediate, which loses water and forms $CH_3CH_2OH$ [65]. The formation of $CH_4$ typically occurs through the formation of the CO$^*$ radical, which is hydrogenated to form the HCO$^*$ radical species followed by subsequent hydrogen addition reactions to give $CH_4$ [66]. The formation of $C_2H_4$ usually

originates from the formation of the $^*$CO–CHO radical via coupling non-adsorbed CO with $^*$CHO. The coupling reaction is followed by subsequent hydrogenation reactions [67]. These different reaction pathways have been shown to depend on the type of TMS used in the electrocatalytic reduction of $CO_2$.

This review will focus on the types of sulfide nanomaterials that have been investigated primarily in the $CO_2$ reduction reactions. There are many reviews on the applications of nanomaterials in electrocatalytic systems. However, most of the work that has been performed in the field does not focus on the specific application of TMS and their composite nanomaterials in the $CO_2$RR. The present review focuses on the application of TMS and their composite nanomaterials in the $CO_2$RR. The theory behind the catalytic activity of TMS is out of the scope of this review. The present review is organized as follows: (1) Introduction, (2) Structure and function of electrocatalysis, Layered rim–edge–plane activities, (3) Overview on the synthesis of TMS, (4) Layered Transition metal sulfide nanomaterials and composite nanomaterials applied to $CO_2$ reduction reactions, (5) Non-layered First row TMS nanomaterials and composite nanomaterials applied to $CO_2$ reduction reactions, (6) Non-layered first row TMS nanomaterials and their composites applied to $CO_2$ reduction reactions, (7) Conclusions/Future Perspectives.

## 2. Structure/Function in Electrocatalysis

Low dimension materials have shown much promise in catalysis and electrocatalysis. The low dimensional TMS have been shown to be catalytically active and have been considered as supports to enhance the reactions. Transition metal sulfides have been widely used as catalysts in various reduction reactions, including hydrogenation reactions [68–75]. $MoS_2$ nanoflowers have been used in the hydrogenation of nitrobenzene [68] and phenanthrene [69]. One of the more classical reactions of TMS, specifically with Co, Ni, $MoS_2$, $WS_2$, and their composites has been the hydrodesulfurization (HDS) reaction [70–75]. In the case of HDS reaction, the catalysts are known to have specific active sites that enhance reactivity by making combinations of Mo with Co or Ni, as well as W with Co and Ni, hence promoting the catalyst activity.

*Layered Rim–Edge–Plane Activities*

The predominant theory behind the reactivity of $MoS_2$, $TiS_2$, and $WS_2$ has been the rim–edge model, in which the basal planes of the material are inactive. The activity of the edge/rim sites of the catalyst has been well studied [76–79]. The Metal and sulfur atoms on the rim or edge are the only species to participate in the reactivity of the catalyst [76–79]. A representation of layered TMS (showing the $MoS_2$ lattice) and the metal- and sulfur-rich edges is shown in Figure 1 [80].

The rim and edge model for the activity indicates that the basal plane of the TMS is non-active or has very low activity [76–79]. From hydrogenation of C–S bonds in the HDS reactions, it has been shown that stacks of 4–5 layers of $MoS_2$ planes show activity. However, the basal plane is inactive towards the reaction where the reactivity occurs at the edge or rim of the $MoS_2$ stack. At the rim sites the active hydrogenation of the C–S bonds occurs, whereas on the edge site the C–S-bond breaking occurs [76–79]. This edge–rim activity observed in HDS reactions is preserved in electrocatalytic reduction reactions of $CO_2$ where the basal plane is inactive and the reactivity is observed only on the rim/edge of the stacked planes. Figure 2 shows the different sites for in the $MoS_2$ rim and edge models and the respective reactions.

The activities on the rim and edges sites translate to the layered TMS, which in turn translate into the $CO_2$ reduction reactions. For the $MoS_2$-, $TiS_2$-, and $WS_2$-based electrocatalysts, the edge sites are far more reactive than the basal planes. However, there has been recent research which has shown creating sulfur deficiencies on the surfaces/basal planes produces reactivity [81–83]. The non-layered TMS typically have the metal located on the surfaces of the crystalline phase and not blocked by the other atoms [84–86]. Figure 3

shows examples of $Fe_3S_4$, $Fe_7S_8$, and CuS crystal structures, which illustrate the availability of the metal ions for reaction [87–89].

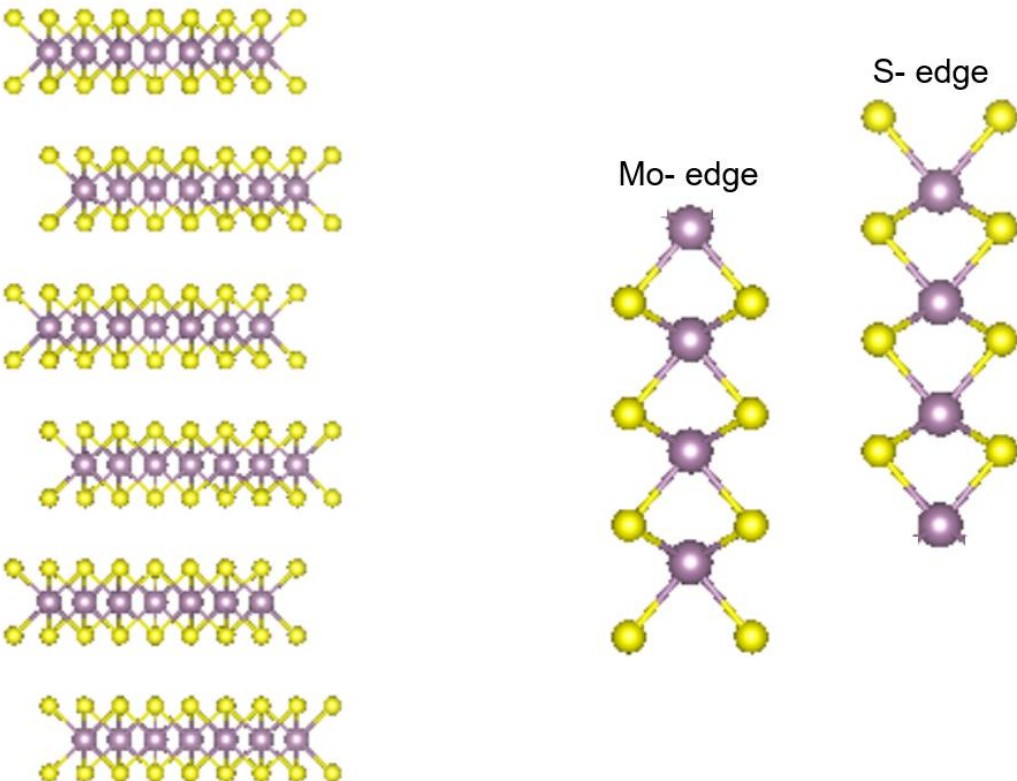

**Figure 1.** Structure of layered TMS dichalcogenides (based on the molybdenite structure), the metal-rich edge sites, and the sulfur-rich edge sites.

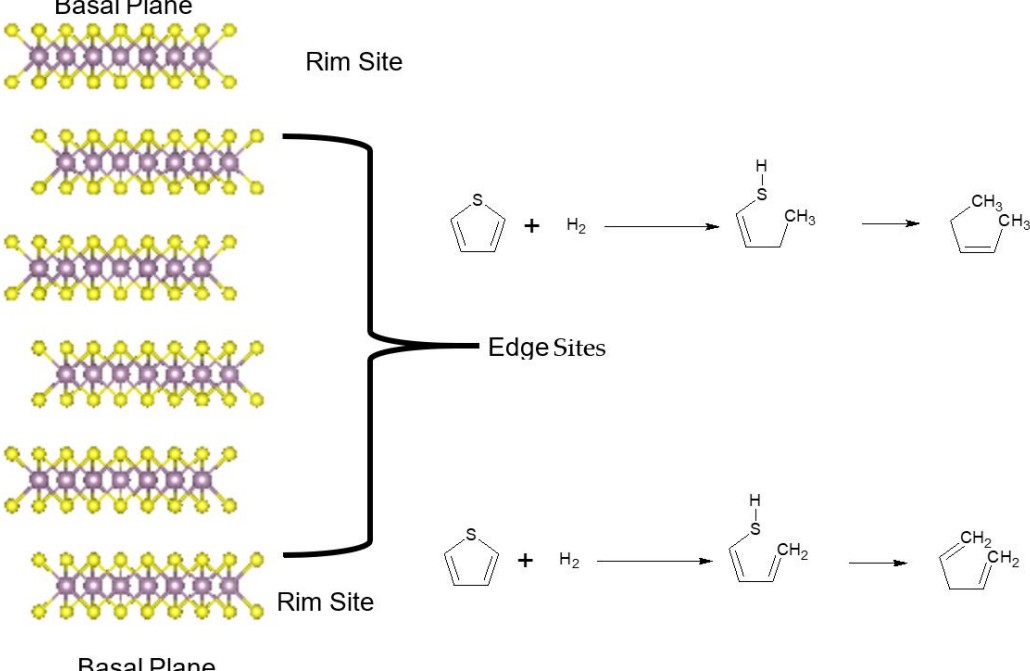

**Figure 2.** Break-down of the layered TMS structure into the basal planes, rim sites, and edge sites, and the HDS reaction associated with the rim and edge sites.

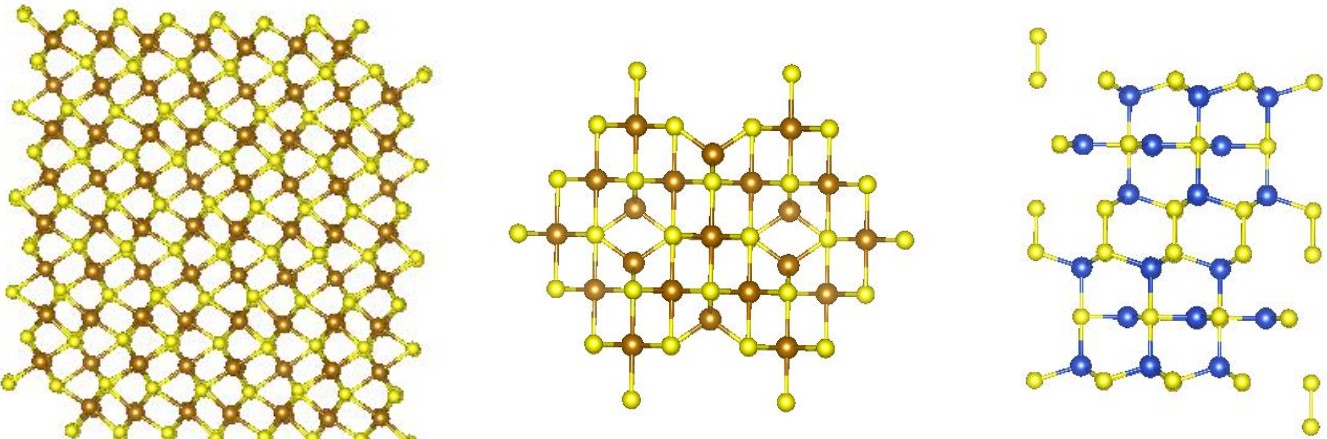

**Figure 3.** Crystal structures of $Fe_3S_4$, $Fe_7S_8$, and CuS.

## 3. Synthesis of Transition-Metal-Sulfide-Based Nanomaterials

The basis of the synthesis of TMS nanomaterials are generally performed using one of the following techniques: low temperature aqueous, hydrothermal, solvothermal, thermal decomposition, or thermal conversion of precursor sulfide compounds [90–95]. Recently, electrochemical synthesis of TMS has become increasingly more popular [96–98]. In addition to the typical solution-based chemical synthesis techniques, solid state reactions have been used in the synthesis of TMS nanomaterials. These techniques include chemical vapor deposition (CVD), physical vapor deposition (PVD), and plasma-based techniques, among others [99–101]. For layered sulfide-based materials, the formation of nanosheets from layered TMS has been achieved through intercalation followed by exfoliation [102–104]. Ball milling has also been used for the synthesis of TMS nanomaterials. The development of supported catalysis, including those used in HDS reactions have been prepared using the insipient wetness technique followed by sulfidation using $H_2S$ in $H_2$ [105]. All these synthesis techniques have applications in electrochemical processes such as batteries, electrodes, electrochemical catalysis, and supercapacitors [106–108].

### 3.1. Low-Temperature Aqueous Reactions

A multitude of TMS has been synthesized using aqueous solutions containing a soluble sulfur source, such as CuS, FeS, and CdS. Typically, this synthesis method leads to the formation of simple sulfide nanomaterials, i.e., single metal ion–sulfide materials. These reactions may be as simple as the addition of $Na_2S$ to a solution followed by the addition of a TM salt. For example, the chemical equation below shows the reaction of copper ions with sodium sulfide in an aqueous solution:

$$S^{2-}_{(aq)} + Cu^{2+}_{(aq)} \rightarrow CuS_{(s)}. \tag{7}$$

This synthesis method has been successful for the generation TMS with simple structures. Due to the low temperature used in the synthesis, the resultant materials from this reaction method are highly amorphous and not well characterized. Silver sulfide nanoparticles have been synthesized using this technique with precipitates sized from 40–50 nm [109]. Lewis has reviewed the precipitation of TMS from aqueous solutions which showed that the precipitation and formation of the TMS nanoparticles depended on several factors, specifically the pH and eH of the solutions [110]. The pH was observed to control the solution speciation of both the sulfur and the metal ions. Moreover, the low temperature aqueous phase reactions became the precursor for other synthesis techniques such as hydrothermal synthesis, which can have more control over phase and generate different phases. Luther et al. used copper sulfide as a model to investigate the low-temperature aqueous solution sulfide precipitates [111]. The results from EPR and NMR showed a

reduction of the $Cu^{2+}$ to $Cu^+$ by the $S^{2-}$, as well as the formation of Cu–S clustered rings in $Cu_3S_3$ solution, which are the basic building blocks for CuS precipitation.

### 3.2. Hydrothermal Synthesis

Similar to the low-temperature aqueous TMS nanomaterial synthesis, a reaction for TM formation is performed using a water-soluble sulfide source, such as $Na_2S$, $(NH_4)_2S$, or NaHS in aqueous solution. The mixture is then transferred to an autoclave and heated to a specific temperature. In these reactions, there is control over the synthesized nanomaterials. Different TMS and TMS-composite nanomaterials have been synthesized using hydrothermal techniques [112–117]. During the hydrothermal synthesis, the crystallite size, structure, and morphology are controlled by both the temperature and the starting materials, as well as the ratios of the materials in the solution mixture.

Li et al. synthesized $MoS_2$ nanowires using $MoO_2$ and $Na_2S$ as the starting materials in an aqueous solution of HCl at 260 °C [112]. Chaudhary et al. have used hydrothermal synthesis techniques to produce $MoS_2$ nanosheets [113]. The authors used a combination of $NaMoO_4$ with thioacetamide and a small amount of sillicontungstic acid (2.8 mM). The mixture was held at a constant 220 °C for 24 h. The results also showed that amorphous nanospheres and nanoflowers were formed at temperatures between 120–150 °C. Zhen et al. synthesized $MoS_2$ using hydrothermal synthesis to generate different morphologies [114]. The results of the study showed that depending on the sulfur and Mo sources and a reductant (either citric acid or ascorbic acid), the morphology and crystallinity of the $MoS_2$ could be controlled while the electrochemical properties could be altered.

Li et al. used hydrothermal synthesis at a temperature of 200 °C for 24 h to generate PbS, CdS, ZnS, CoS, and $Cu_2S$ nanoparticles using 1-butyl-3-methlyimidazole thiocyanate as the sulfur source [115]. The authors used the chloride of the respective metals as the metal ion source for the sulfide. The results showed the formation of dendrites by PbS and CdS while the other metals formed spherical particles. Chen and Fan used hydrothermal synthesis to make $NiS_2$, $CoS_2$, $FeS_2$, $NiSe2$, $MoS_2$ and $MoSe_2$ at the nanoscale size [116]. The authors used $Na_2S_2O_3$ with the chloride salts of $Ni^{2+}$, $Co^{2+}$, and $Fe^{2+}$ metal ions over a temperature range between 140 and 150 °C for 12 h, which resulted in the pyrite type structure [116]. $NiSe_2$ was synthesized using $NiCl_2$ and $Na_2SeSO_3$, which resulted in the formation of $NiSe_2$ nanocrystals. The synthesis of $MoS_2$ and $MoSe_2$ was successful with $Na_2MoO_4$ and $Na_2S_2O_3$ or $Na_2SeSO_3$ which resulted in the formation of a hexagonal layered structure after annealing in an inert atmosphere at 350 °C. Alzaid et al. synthesized NiMnS at different ratios using hydrothermal synthesis for use as electrodes in energy storage applications [117]. The authors used metal chlorides in the presence of sodium sulfide at a reaction temperature of 170 °C for 1 day which resulted in the formation of mixed metal TMS with high specific capacity and rate performance. The carbon hybrid delivered an energy density of 46 Wh/kg.

### 3.3. Thermal Decomposition of a Sulfur Source

One of the more traditional methods for the synthesis of $MoS_2$ nanoparticles and TMS–$MoS_2$ composite nanomaterials is through the thermal decomposition of ammonium tetrathiolmolybdate (ATM). This method is commonly applied to the synthesis of ammonium tetrathriotungstate (ATT) [118,119]. ATM can be formed in a basic aqueous solution by the reaction of $(NH_4)_2S$ or $H_2S$ with ammonium molybdate as shown below:

$$4S^{2-} + MoO_4{}^{2-} \rightarrow MoS_4{}^{2-}. \tag{8}$$

The ATM can be crystallized or further reacted with TM depending on the desired product such as the generation of $CoMoS_2$, $NiMoS_2$, or $MoS_2$ [120–122]. For $MoS_2$ nanomaterials, the synthesized ATM is crystallized and placed into a tube furnace with an inner atmosphere containing $H_2$ gas [122]. The furnace is then heated to the desired temperature, typically 350 °C or above, which can be used to control the crystallinity of the nanomaterial.

In fact, higher temperatures result in much more stacking in the TMS. The heated sample goes through the chemical reaction shown below:

$$MoS_4{}^{2-} + H2(g) \rightarrow MoS_{2(s)} + H_2S_{(g)}. \tag{9}$$

An alternative synthesis method reported recently showed that elemental sulfur can be used to synthesize $CoMoS_2$ [72], while $MoS_2$ was synthesized by precipitating CoMoO4 on top of elemental sulfur through a reflux synthesis. The $CoMoO_4/S$ material was converted thermally using $Ar/H_2$ (90:10) at 350 °C. The final material synthesized through this method was the $Co_9S_8$–$MoS_2$ HDS catalytic material.

Han and Gao synthesized $Fe_7S_8$ and $Fe_3S_4$ nanosheets using single source precursors [123]. The authors prepared Fe(diethyldithiocarbamate)$_2$(1,10-phenanthroline) and Fe(diethyldithiocarbamate)$_3$ using solvothermal synthesis in oleylamine. The Fe(diethyldithiocarbamate)$_2$(1,10-phenanthroline)-based precursors were decomposed at temperatures between 240 and 320 °C, which showed that at 260 °C, the product was identified as monoclinic $Fe_7S_8$. At 320 °C, the product exhibited diffraction features of hexagonal troilite FeS. When the Fe(diethyldithiocarbamate)$_3$ complex was decomposed, the sample showed the presence of $Fe_3S_4$. Allonso et al. decomposed ammonium thio salts for the synthesis of $MoS_2$ and $WS_2$ catalysis [69]. The authors prepared ATT and ATM materials and pressed each at 350 and 700 MPa while decomposing the catalysts at 400 °C, in a $H_2S/H_2$ atmosphere. The higher pressures resulted in a catalyst with higher surface area.

*3.4. Solvothermal Processes*

Solvothermal processes are typically used in two distinct cases, when the starting materials are reactive in water or very oxyphilic, such as titanium or vanadium chlorides or other high oxidation state transition metals; these materials can be used as the starting TM in the solvothermal process, which can react to give the metal oxide or hydroxide. Alternatively, when the synthesis temperature cannot be achieved using water, due to pressure constraints or temperature constraints of the reaction vessels, then alternative methods should be used to form the product. Large organic molecules give the advantage of high boiling points with low vapor pressure. An example of the limitation of water as a solvent at temperature is that, when heated to 300 °C it has a vapor pressure of 85 atm, and after heating to 360 °C water has a vapor pressure of approximately 184 atm. The vapor pressure of water becomes a limiting factor and alterative solvents are to be explored for high temperature syntheses. In addition, the solvothermal process allows for the control of the phase, particle size, and morphology [124–135].

Titanium sulfide nanomaterials have been synthesized using solvothermal processes using titanium tetrachloride with elemental sulfur in 1-octadecene at 300 °C in one hour of reaction [124]. Nickel sulfides in the pyrite phase, $NiS_2$ and NiS, were successfully synthesized using the solvothermal process using ethanol and ethylenediamine as solvents, respectively [125,126].

Many of the iron sulfides have been synthesized using solvothermal processes, for example, Kar and Chaudhuri synthesized iron sulfide nanowires using ethylenediamine as the solvent at 180 °C for 12 h [127]. Cantu et al. used a mixture of ethylene glycol and water to synthesize an $Fe_7S_8$ nanomaterial at 180 °C in 1 h [128]. Xufeng et al. used solvothermal processes for the controlled synthesis of pyrite in ethylenediamine at 130 °C [129]. Zhang and Chen et al. showed the successful synthesis of $Fe_3S_4$ using ethylene glycol with iron(III) chloride and thiourea at 180 °C [130]. Copper sulfides and $MnS_2$ were successfully synthesized using solvothermal processes with ethanol as the solvent and thioacetamide as the sulfiding agent [131]. Goria et al. have used ethylenediamine as the solvent at a temperature 130 °C for 12 h and phase-pure $Cu_2S$ was synthesized, which showed a dendritic-type structure [132]. While the rection at 130 °C for 4–8 h showed a similar type of morphology. The study showed that heating the reaction mixture for 8 h at 130 °C resulted in $Cu_{31}S_{16}$. Coa et al. used a solvothermal process which resulted in 1T-2H $MoS_2$ nanoflowers with defects [133]. The authors used ammonium thiocyanate as the sulfur

source and ammonium molybdate with a mixture of ethane/water/glycerin as the solvent. Xu et al. performed solvothermal syntheses using ethylenediamine and dodecanthiol to form CdS nanowires [134]. The authors successfully synthesized CdS nanowires at 180 °C, with diameters of 25 nm and lengths into the µm ranges, which showed controlled synthesis of CdS nanomaterials. Valdes et al. on the other hand, synthesized nanosized $CoMoS_2$ doped with La ions for the desulfurization of fuels [135]. The authors prepared a $CoMoS_4$ precursor which was doped with different amounts of La ions to replace Co atoms from 5 to 25%. The precursor was converted to the sulfide in decalin at 350 °C in a sealed autoclave reactor. The catalysts showed high activity and some resistance to carbonization.

### 3.5. Electrochemical Synthesis

Recently, the electrochemical production of different catalyses, such as the TMS have become increasingly popular as the modification of the TMS surfaces to generate sulfur deficiencies in the surfaces was applied to generate catalytically active materials [96–98,136–138]. Typically, an electrochemical synthesis is performed in a solution using cyclic voltammetry between specific voltages. Fotouhi et al. successfully synthesized $Cu_2S$ nanoparticles from aqueous solutions of $Na_2S$, PVP (added as a stabilizer), and a $KNO_3$ electrolyte solution; a sacrificial Cu anode was converted to $Cu_2S$ [96]. The results showed that the median distribution of the nanoparticles was between 12 and 17 nm and the material was in the $Cu_2S$ phase. Shamsipur et al. reported results on the electrochemical synthesis of CuS nanoparticles [97]. The authors used a two-electrode system with platinum and sacrificial copper electrodes. The results showed that as the voltage in the solution was increased, the production of CuS was decreased. In addition, the size and shape of the nanoparticles were dependent on the concentration of sulfide in solution. Frazli et al. synthesized NiS nanoparticles using electrochemical synthesis techniques [98]. The study showed that NiS nanoparticles with spherical shapes with sizes ranging from 17 to 27 nm could be synthesized. The authors used voltages from 5 to 15 V with a sacrificial Ni electrode in a solution containing sodium sulfide and as the electrolyte. Zhang et al. synthesized a NiCoS nanomaterial for supercapacitors using electrochemical processes [136]. The authors prepared a Ni–graphite foam from which the Ni was dissolved to give a graphite foam. The graphite foam had the Ni–CoS deposited electrochemically from solution using cyclic voltammetry cycling from −1.2 to 0.2 V. The preparation was formed using a solution of thiourea and both $NiCl_2$ and $CoCl_2$; the pH was adjusted using ammonium solution. Shankar et al. showed a one-step electrochemical synthesis process to generate various TMS, including Fe, Ni, and Cu supported on a bare Ni foam [137]. The authors performed five cycles of applied voltages from 1.63 V to −0.16 V to an electrolyte solution with thiourea as the sulfur source. The results showed average grain sizes of ∼326 nm for FeS, ∼254 nm for CuS (with a spherical morphology), and ~326 nm for the NiS and CoS nanoparticles. The FeS showed the highest current densities even higher than $RuO_2$. Golpalakrishnan et al. synthesized $MoS_2$ quantum dots using electrochemical techniques [138]. The authors performed an electrochemical etching of bulk $MoS_2$ to generate nanoparticles ranging in size from 2.5 to 6 nm, which had photoluminescence properties. The results showed that in an aqueous ionic liquid solution of 1-butyl-3-methylimidazolium chloride and lithium bis-trifluoromethylsulphonylimide with 5 V applied across the $MoS_2$ quantum dots could be grown.

### 3.6. Chemical Vapor Deposition (CVD)/Physical Vapor Deposition (PVD)/Plasma Synthesis

CVD and PVD techniques as well as plasma techniques were utilized for the synthesis of TMS for multiple purposes. PVD consists of vaporizing a solid for deposition under vacuum, which showed to have great control in synthesis. While CVD is a vapor deposition technique from a thin film, the two techniques are very similar, however, the major difference is the deposition material in PVD is a solid and in CVD, it is a gaseous molecule. Additionally, the temperatures used in CVD are typically higher than PVD.

Plasma deposition is generally a process defined as the deposition of a material from the action of a plasma.

Farwa et al. synthesized a bimetallic $Ni_3S_2/MnS_2$ composite nanomaterial using PVD [99]. The authors used metal nitrates in conjunction with $CS_2$, in a two-step synthesis process. A thiocarbamate precursor was synthesized and decomposed in a tube furnace in an inert atmosphere and $Ni_3S_2/MnS_2$ nanoparticles were synthesized. Ge et al. utilized atmospheric pressure CVD to synthesize CdS, ZnS, $Cu_{7.2}S_4$, NiS, CoS, $Fe_7S_8$, MnS, $Cr_2S_3$, and $WS_2$ nanowires [100]. The nanoparticles synthesized through the CVD process averaged between 20 and 30 nm on the low end to 100–150 nm on the high end and ranged in length from 1 to 40 μm. Wang et al. used CVD to deposit 2-D TMS nanoparticles on carbon paper for electrocatalytic hydrogen evolution [139]. The authors grew $MoS_2$, $NbS_2$, and $WS_2$ nanosheets, where $WS_2$ showed the highest HER activity with an average size between 100 and 200 nm. The $MoS_2$ had an average size of 500 nm while the $NbS_2$ had an average size of 500 nm. Royki and Hirotatsu used CVD to synthesize NiS nanoplates, the authors used diethyl amine and $CS_2$ which synthesized a Ni–dithiocarbamate complex. The compound was decomposed in an inert atmosphere at 400 °C to generate NiS [140]. Zhai et al. used a CVD technique to synthesize CdS nanorods [141]. The authors used a Cd dithiocarbonate which was decomposed in a tube furnace at 450 °C in nitrogen. The CdS nanoparticles had a tetrapodal geometry with an average size between 4 and 5 nm. Khan et al. synthesized phase-pure CuS nanostructures using a CVD technique [142]. The authors used a single precursor Bis(O-isobutyldithiocarbonato)copper(II) complex [$Cu(SCSOCH_2CH(CH_3)_2)_2$] and decomposed the material at temperatures from 250 to 350 °C. The CuS formed at 250 °C was spherical in morphology, at 300 °C, the nanoparticles had plate-like morphology, and at 350 °C the authors noted a flower morphology.

Zheng et al. used non-thermal plasma deposition for the synthesis of a $MoO_3$@$MoS_2$-CuS composite nanomaterial for light harvesting [101]. The authors used sodium molybdate in conjunction with $MoO_3$ nanorods and ultrasonic treatment. The authors added $MoO_3$ and $Cu(NO_3)_2$ and dried and placed them in a plasma to be sulfided using $H_2S$/Ar at 100 W for 40 min. The results showed the formation of nanomaterials with an average grain size of 50 nm for the $MoS_2$ and micron sized $MoO_3$ particles. The CuS size was not discussed. Zhao et al. used cold plasma (non-thermal plasma) to synthesize Cr-doped ZnS catalysts for $H_2S$ decomposition [143]. The authors prepared an impregnated $Al_2O_3$ support with Zn and Cr in an aqueous solution to prepare the precursor, which was calcined at 450 °C in air. The calcined sample was then sulfurized in a cold plasma (400 °C) using a mixture of (Ar:$H_2S$ 90%:10%) resulting in 8 nm nanoparticles. Basuvalingam et al. used low-temperature plasma for the synthesis of 2D TMS, $TiS_2$, and $TiS_3$ nanolayers [144]. The plasma system allowed for the atomic layer by layer deposition of $TiS_2$ and $TiS_3$ nanolayers. The authors used a $H_2S$-based plasma between 150 and 200 °C. The $TiS_2$ and $TiS_3$ materials showed electrical transport properties and photoluminescence properties, respectively.

## 4. Layered Transition Metal Sulfide Nanomaterials and Composite Nanomaterials Applied to $CO_2$ Reduction Reactions

### 4.1. $MoS_2$

As mentioned earlier, $MoS_2$ is a well-known catalyst in different fields including electrocatalysis. Siahrostami et al. reported that the edge sites of $MoS_2$ have the catalytic capability for $CO_2$ reduction to form CO [145]. The authors demonstrated that the bridging S atom at the edges of $MoS_2$ could selectively bind the intermediate COOH* over the CO product, which gave rise to the transition-metal scaling relationship of intermediates in $CO_2$ reduction processes and therefore remarkably enhanced $CO_2$ reduction activities on transition-metal catalysts. Francis et al. investigated the conversion of $CO_2$ to 1-propanol at $MoS_2$ electrodes [146]. The results showed that at a potential of $-0.59$ V (compared to a standard hydrogen electrode (SHE)) for large $MoS_2$ crystals, an efficiency of approximately

3.5% was observed, and with thin films it showed approximately 1%. However, the bulk crystals showed degradation by loss of S as $H_2S$ but the formation of $H_2S$ was not observed in the thin film studies.

Lv et al. investigated the electroreduction of $CO_2$ to CO using $MoS_2$ and N-doped C dots [147]. In that work, the $MoS_2$-N-doped carbon dot hybrid was prepared using a solvothermal process, with the formation of the C dots on the $MoS_2$ nanosheet surfaces in dimethyl formamide (DMF). The composite catalysts showed a faradaic efficiency of 90% with an overpotential of 0.130 V. The authors also performed density functional theory (DFT) calculations which indicated that N-doping of the $MoS_2$ could decrease the energy barrier of the *COOH intermediate and generate more electrons on the Mo edge of the $MoS_2$ and thus enhance the catalytic activity. Lv et al. investigated the effects of decorating E-$MoS_2$ nanosheets with flurosilane (FAS) for the reduction of $CO_2$ to CO [148]. The authors exfoliated $MoS_2$ sheets from the bulk using ball milling and subsequently decorated the surface with FAS. The FAS-modified $MoS_2$ sheets showed the ability to be tuned for syngas production from $CO_2$ and $H_2O$ mixtures. Additionally, the FAS-modified $MoS_2$ sheets showed a faradaic efficiency for CO production of 81% at a current density of 61 mA and an over-potential of −1.1 V. DFT calculations indicated that the FAS modified the electrochemistry of the edge Mo sites which facilitated the desorption of the CO and thus enhanced reaction over non-modified $MoS_2$ sheets.

Qi et al. investigated the layering of $TiO_2$ over $MoS_2$ nanosheet arrays for the electrocatalytic reduction of $CO_2$ to ethanol [149]. The $TiO_2$–$MoS_2$ composites were formed through atomic layer deposition. The composite material showed a 50% faradaic efficiency at −0.6 V. The authors also performed DFT computation studies of the electrocatalysis, which indicated the formation of a Mo–Ti active site at the interface between the $TiO_2$ and $MoS_2$. Yu et al. investigated the electrocatalytic reduction of $CO_2$ to CO using $TiO_2$-modified $MoS_2$ [150]. The optimized composite in $KHCO_3$ solution showed an onset over-potential of 100 mV and a maximum faradaic efficiency of 82% at -0.7 V with a current density of 68 mA/cm$^2$. The authors also performed DFT calculations which indicated that the Ti–S bonds formed between the $TiO_2$ and $MoS_2$ composite changed the electric properties of the $MoS_2$ and adsorption on the Mo sites. Li et al. investigated $MoS_2$ rods supported on $TiO_2$ NPs for the electrocatalytic and photo-enhanced electrocatalytic reduction of $CO_2$ to methanol [151]. The electrocatalytic reduction showed a faradaic efficiency of 42% percent; however, the addition of light showed a faradaic efficiency of 112%. The authors concluded that the mechanism for the photo-enhancement was from the photo-oxidation of water at both the anode and cathode surfaces.

Peng et al. performed Co doping of $MoS_2$ nanoparticles, which were applied to the electrocatalytic reduction of $CO_2$ to form methanol [152]. The authors synthesized a material with an average size of 30 nm, which gave a 35 mmol/L of methanol after 350 min of reaction. In addition, the over-potential for the $CO_2$ reduction was observed to be −0.64 V. Hussain et al. synthesized Cu on graphitic g–$C_3N_4$/$MoS_2$ composites for the reduction of $CO_2$ to alcohols [153]. The Cu–g–$C_3N_4$/$MoS_2$ composite showed faradaic efficiencies of 19.7 and 4.8% for the formation of methanol and ethanol, respectively. The faradaic efficiency was better than that observed for Cu–g$C_3N_4$ and Cu$MoS_2$ alone. The material also showed good stability to hold a constant current density over 30 h of reaction. Abbasi et al. investigated the effect of the edge structure of $MoS_2$ in electrocatalytic reduction of $CO_2$ [154]. The authors used an ionic liquid and showed that a 5% dopant of $NbS_2$ showed a 10× enhancement of the reduction of $CO_2$ over pristine $MoS_2$. The results showed the over-potential working range was between 0.05 and 0.150 V in 1-ethyl-3-methylimidazolium tetrafluoroborate as the ionic liquid with CO as the product of the reaction.

Hussain et al. investigated a CuO–ZnO–$MoS_2$ composites material for $CO_2$ reduction to make alcohols [155]. The $MoS_2$ was used as a support for the CuO–ZnO materials and

provided a synergistic effect on the electrocatalytic properties of the CuO–ZnO system. The composite showed excellent performance in the reduction of $CO_2$ into methanol, which showed approximately 25% conversion, with a current density of 17.3 mA/cm$^2$ at a potential of $-1.3$ V (compared to Ag/AgCl). Shi et al. studied Cu nanoparticles inter-dispersed with $MoS_2$ nanoflowers for the electrocatalytic reduction of $CO_2$ [156]. In that study a microwave-assisted synthesis was used to produce the nanoparticles, which had an average size between 5 and 20 nm [156]. The composite material was successful in producing CO, $CH_4$, and $C_2H_4$. The $MoS_2$ nanoflowers in the absence of copper nanoparticles were still active but at a strongly reduced rate of approximately 1/7 time.

The results discussed above show great progress for the application of $MoS_2$ and $MoS_2$-based composite nanomaterials in $CO_2$ reduction reactions. However, the applications that show the most promise of providing valued-added materials appear for the most part to utilize $MoS_2$ as a base support for a secondary material. In the following section, we review existing modeling (simulation) results reported on the use of $MoS_2$ and $MoS_2$-based composite nanomaterials for $CO_2RR$

*4.2. Computational Studies Using MoS$_2$*

The results from simulations have also been reported on for the reduction of $CO_2$ using different metal sulfides with the aim to provide more insights into its mechanisms, efficacy, and efficiency. Xie et al. studied the mechanism of reduction of $CO_2$ to CO using $MoS_2$ monolayers as catalysts [157]. The simulation results were performed using DFT calculations. The results indicated that the Mo exposed on the edges of the $MoS_2$ were the adsorption sites for the $CO_2$ and the O=C=O bonds which were reconfigured during the adsorption process between two Mo centers. In addition, the first $H^+$ reduction reaction occurred through a different pathway which pushed product selectivity towards the formation of CO. Mao et al. on the other hand, investigated the effect of modification of the $MoS_2$ edge structure by the addition of transition metals on electrocatalytic $CO_2$ reduced to CO using DFT calculations [158]. The focus of the study was on the insertion of V, HF, and Zr into edge sites of the $MoS_2$. The simulation or (modeling) results indicated that the activity did not depend on dopant concentration. However, locating the dopant close to the Mo active site was extremely important for desorption. Similarly, Datar et al., using DFT calculations, investigated the reduction reaction of $CO_2$ on $MoS_2$ monolayers [159]. In that work, the authors investigated the formation of 2H, 1T, and 1T' phases in $MoS_2$ supported on Ag, Au, and Cu. The results indicated that the $MoS_2$ basal planes were relatively unaffected by the supports, but the Mo active sites were poisoned by the support. Additionally, the support effects were important and needed to be considered.

Yu et al. used computational methods to investigate single-atom catalysts based on first-row transition elements on $MoS_2$ for the electrocatalytic reduction of $CO_2$ to methanol [160]. The results showed that the Ni–$MoS_2$ combination was the most stable catalyst of the series using electronic structure studies. Ren et al. used DFT computational methods to investigate single-atom-supported catalysts on $MoS_2$ for the electrocatalytic reduction of $CO_2$ [161]. The authors investigated Pd, Co, No, and Pt metals supported on $MoS_2$, which were shown to be efficient in the electrocatalytic reduction of $CO_2$. The Fe, Co, Ni, and Pt on $MoS_2$ produced primarily $CH_4$ and had a low limiting potential. In addition, the formation of HCOO$^*$ was determined to be one of the key reactive descriptors for the reaction. The authors also noted that the $MoS_2$, $Ws_2$, and $WSe_2$ systems with single-atom catalysts showed promise. Li et al. on the other hand used DFT calculations to explore the effects of edge-exposed $MoS_2$ hybridized with N-doped carbon [162]. The results in that study showed that a high number of exposed edge sites on the $MoS_2$ provided a high number of active sites, which resulted in lowering the onset potential. In addition, the results showed a 93% faradaic efficiency at an over-potential of 0.59 V. CO was the major product formed in the reaction [162].

The modeling results discussed above are of great interest, as a new mechanism for $CO_2$ reduction was explored, which is difficult to be studied experimentally. These computational studies really supplemented and helped explain the reaction mechanisms. However, more work needs to be done on other layered TMS. For example, $TiS_2$ has been also used for $CO_2$ reduction and more importantly, both modeling and experimental results were reported on its effect on the electrocatalysis reduction of $CO_2$.

### 4.3. Experimental Studies on $CO_2RR$ Using $TiS_2$

Another layered transition metal dichalcogenide investigated in for the conversion of $CO_2$ has been $TiS_2$. Aljabour, investigated the reduction of $CO_2$ to CO using $TiS_2$, which showed a faradaic yield of 64% at a potential of 0.4 V and current density of 0.5 mA/cm$^2$ [163]. The driving mechanism of the $CO_2$ reduction was determined to be the $CO_2$ binding to the active disulfide planes. In a similar study, Aljabour et al. investigated the active sulfur sites in semi-metallic $TiS_2$ which enabled the electroreduction of $CO_2$ [164]. The experiments were performed using in situ FTIR where the results showed that the $CO_2$ was bound to sulfur to form monothiocarbonate species. The monothiocarbonate species can steer the reduction reaction towards the formation of CO [164]. The results reported in that work showed cathodic efficiencies of up to 54% at a current density of 5 mA/cm$^2$.

### 4.4. W-Based Materials

$WS_2$ is a great catalytic material but there is not much work reported in the literature on the use of $WS_2$ in the electrocatalytic reduction of $CO_2$. However, Asadi et al. found that $WSe_2$ was an effective catalyst in the reduction of $CO_2$ in ion liquid media [165]. The results showed that $CO_2$ could be reduced to CO with a faradic efficiency of 24% with a TOF of 0.28 s$^{-1}$. In addition, the reaction occurred at a low over-potential of 54 mV in 50 vol% EMIM–BF4 in water.

### 4.5. $WS_2$ Computational Studies

Tong et al. studied the effects of engineering the edge layers of $WS_2$ on the electrocatalytic reduction of $CO_2$ using DFT calculations [166]. The aim of the modeling work was to investigate the doping of the $WS_2$ structure with Zn, Fe, Co, and Ni, which showed a change in the bond strength between the W and S atoms and increased electrical conductivity [166]. In addition, the catalytic activities of the composite materials could be increased, and an optimum catalyst was observed in the calculations using Zn as the dopant with an over-potential of $-0.51$ V. Fonseca et al. on the other hand, used DFT calculations to study the effect of Fe doping on $WS_2$ [167]. The results showed that the Fe has a preferential binding to two sulfur atoms on the edge site and showed breaking of the W–S bonds at the edges of the nanomaterial. The results also showed that $CO_2$ was activated on the $WS_2$ with bond angles of 129°, and the binding occurred though physisorption. These computational studies performed on $WS_2$ do provide some mechanistic insight into the $CO_2RR$ using $WS_2$-based nanomaterials. These insights and mechanisms observed with $WS_2$ are very similar to those in $MoS_2$-based materials. These insights include modeling the rim–edge and distortions of the O=C=O bond angles after adsorption, and the importance of the metal sites in the reaction mechanism.

Table 1 shows a summary of the different products generated using layered $MoS_2$, $TiS_2$, $WSe_2$ and their respective nanocomposites. As can be illustrated in the table, these nanomaterials are very versatile for their applications in the $CO_2RR$. The majority of the common small molecule products (CO, $CH_3$, $CH_3OH$, $CH_3CH_2OH$, and $C_2H_4$) observed in the $CO_2RR$ were formed using $MoS_2$, $TiS_2$, $WSe_2$ and their respective nanocomposites. However, formate was not shown as a reaction product but was a postulated product using computation.

**Table 1.** Summary of products in the $CO_2RR$ using the various TMS-layered nanomaterials and their associated composite nanomaterials.

| $CO_2RR$ Product | Material | Reference |
|---|---|---|
| CO, $CH_3CH_2CH_2OH$ | $MoS_2$ | [145,146] |
| CO | $MoS_2$–$C_3N_4$ | [147] |
| CO | $MoS_2$ exfoliated–flurosilane capped | [148] |
| $CH_3CH_2OH$, $CH_3OH$, CO | $TiO_2$–$MoS_2$ | [149–151] |
| $CH_3OH$ | Co–$MOS_2$ | [152] |
| $CH_3CH_2OH$, $CH_3OH$ | Cu–$C_3N_4$–$MoS_2$ | [153] |
| CO | $NbS_2$–$MoS_2$/ionic liquid | [154] |
| $CH_3OH$ | CuO–ZnO–$MoS_2$ | [155] |
| CO, $CH_4$, $C_2H_4$ | Cu–$MoS_2$ | [156] |
| CO | $TiS_2$ | [163,164] |
| CO | $WSe_2$ | [165] |

## 5. Non-Layered First Row TMS Nanomaterials and Composite Nanomaterials Applied to $CO_2$ Reduction Reactions

### 5.1. $CuS_x$-Based Nanomaterials

Lim et al. investigated the application of $CuS_x$ in the electrochemical reduction of $CO_2$ to formate [168]. The authors used industrial $CO_2$, which contained $H_2S$ and performed an in situ formation of $CuS_x$ from a Cu foil. The study showed that the average size of the $CuS_x$ nanoparticles was 133.2 nm and the nanoparticles had a faradaic efficiency of up to 72% at −6 V and showed stability for up to 72 h of operation. Philips et al. investigated the deposition of CuS on Cu foils for the electrocatalytic reduction of $CO_2$ to formate [169]. In that study, CuS was deposited on Cu foil using electrodeposition to prepare a nanowire of the CuS. The products of the reduction included hydrogen at low potential, then CO and formic acid at intermediate over-potentials, and $CH_4$ at a high over-potential. The faradaic efficiency of the CuS on Cu foils was higher than of the foil alone. Chen et al. reported results on the investigation of copper sulfide compound properties in the electrochemical reduction of $CO_2$ [170]. The material was synthesized through electrochemical deposition of CuS. The results of the study showed that, as particle size increased, better faradaic efficiencies were observed, up to approximately 70%. The results showed low production of CO, with formic acid formation preferred at intermediate current densities of between −1 and −10 $mA/cm^2$. Oversteeg et al. investigated $Cu_2S$ nanoparticles on carbon for the electrochemical reduction of $CO_2$ and the formation of formate [171]. The authors prepared the CuS and $Cu_2S$ nanoparticles through liquid phase sulfidation of CuO on a carbon substrate. The results showed that with less than 4% of the carbon substrate covered, the reaction was selective towards the formation of formate at low current densities [171]. In general, the $Cu_2S$ showed higher faradaic efficiencies over the CuS and the CuO nanoparticles at current densities of −1.5, −3, and −4.5 $mA/cm^2$.

Hu et al. used FTIR to investigate the formation of formate by the deposition of copper sulfide on nitride graphene [172]. The synthesized CuS on nitride graphene ($C_3N_4$) showed high faradaic efficiencies of 82% at an over-potential of 57 mV for the formate formation. The FTIR results showed that one of the oxygen atoms in the $CO_2$ was bound to the $Cu_2S/C_3N_4$ to form OCHO[*] species. The authors also used DFT calculations which showed that the sulfur promoted the activation of the interfacial water formation H species. He et al. investigated the use of $CoS_2$ as a template for the generation of $Cu_2S$ as a $CO_2$ reduction catalyst for the generation of formate [173]. The authors used electrochemically driven cation exchange to replace the Co to generate the $Cu_2S$ while maintaining the morphology of the $CoS_2$ nanoparticles. The copper-based catalysts showed a faradaic efficiency of 87% for the generation of formate.

Deng et al. studied the selective electrochemical reduction of $CO_2$ to formate on $CuS_x$ catalysts [174]. The authors prepared electrochemically deposited $CuS_x$ catalysts that could reduce $CO_2$ to produce formate. The faradaic efficiency of the nanomaterial was 75% with a current density of −9.0 $mA/cm^2$ at an over-potential of −0.9 V. The

formation of other products was suppressed to a faradaic efficiency of 4%. Based on Raman spectroscopy experimental results, the authors determined that the HCOO* radical was the major intermediate. The authors also concluded that the selectivity of the catalyst towards formate was due to the S doping in the Cu matrix which caused the formation of the HCOO* and not the COOH* intermediate species. Zhao et al. investigated the electrocatalytic reduction of $CO_2$ to $CH_4$ on CuS nanosheet arrays [175]. The authors synthesized CuS nanosheets supported on Ni foam. The study showed excellent faradaic efficiency of 73% for the formation of $CH_4$, with stability for up to 60 h of reaction time. The formate was formed during the reaction; however, it was a minor product, as was the formation of CO, while $H_2$ was formed at higher over-potentials between −1.3 and 1.25 V. At over-potentials of −1.2 to −0.8 V, $CH_4$ was the preferred product of the reaction and above −0.8 V, $H_2$ became the major product [175]. Shinagawa et al. conducted experiments to investigate the sulfur modification of Cu catalysts for the electrocatalytic reduction of $CO_2$ to form formate [176]. The CuS was synthesized through modification of Cu electrocatalysts with sulfur forming CuS nanoparticles. The study showed a positive correlation between particle size and selectivity towards the formation of formate, as the nanoparticle size increased so did the amount of formate that was formed. The results in that study showed a maximum faradaic efficiency of 80% at an over-potential of −0.8 V [176]. Other products in the reaction mixture were also observed, including CO, $CH_4$, $C_2H_4$, CO, and $H_2$, but formate was the major product [176].

The results reported on the use of CuS as an electrocatalyst showed great diversity in the products resulting from the $CO_2$RR. Within the CuS/$Cu_x$S nanomaterials, a higher diversity of products is observed compared to the $MoS_2$, $WS_2$, $TiS_2$, and the $Bi_2S_3$ materials. The higher diversity of products indicates that the CuS/$Cu_x$S materials provide more pathways for the $CO_2$RR.

*5.2. $NiS_x$-Based Nanomaterials*

Han et al. investigated the electrochemical reduction of $CO_2$ to CO on NiS nanoparticles [177]. XAS experiments were used to investigate the behavior of the catalysts during the process, where a structural change in the NiS was observed [177]. The authors noted that the change in the Ni–S structure was similar to the that of single-atom-N-coordinated Ni. In addition, the results showed that the O–S bond formation was modulated at a current density of 0.3 mA/cm$^2$ [177]. However, the catalyst showed low stability due to the loss of S atoms. The faradic efficiency observed for the NiS was approximately 85%.

Zhao et al. used $FeS_2$ on NiS to reduce $CO_2$ to methanol which was synthesized through a hydrothermal process [178]. The study showed that the reaction was initiated at an over-potential of 0.280 V; however, a faradic efficiency of 64% at a potential of −0.6 V (reversible hydrogen electrode) was observed [178]. The catalyst showed stability for up to 4 h which was attributed to the ladder structure of the nanocomposite. In addition, the results indicated that the active sites were located between the interface of the $FeS_2$ and NiS. Yamaguchi et al. investigated the reduction of $CO_2$ on nickel containing iron sulfides as a model for hydrothermal vents where $CO_2$ reduction was observed [179]. The authors synthesized $FeNi_2S_4$ using hydrothermal synthesis as violarite in its mineral form. The results showed that the pristine greigite was ineffective at reducing $CO_2$; however, that of Ni-substituted greigite was improved by approximately 85%. In addition, methane was the preferred product with the Fe/Ni nanoparticles, while hydrogen was the preferred product with pure greigite. Zhang et al. studied the synthesis of Ni–S–C nanoparticles from S-doped Ni–triazolate starting material for the $CO_2$ reduction to CO [180]. The authors synthesized a pure Ni(II)–triazolate complex as a starting material followed by pyrolysis at three temperatures, 800, 900, and 1000 °C, which resulted in a $Ni_3S_2$ material with carbon. The highest faradic efficiencies were observed at −1.5 V (vs. Ag/AgCl). The results showed faradic efficiencies of approximately 67%, 45%, and 40% at −1.5 V for the 800, 900, and 1000 °C synthesized products, respectively. The results reported on the use of nickel-sulfide-based nanomaterials in the $CO_2$RR are of great interest. The hybrid nanomaterials

appear to show higher activities than the individual $Ni_yS_x$ materials. The Fe doping of the $Ni_yS_x$ and the Ni doping of the $Fe_yS_x$ nanomaterials do enhance the reactivity of the $Ni_yS_x$. There appears to be some type of synergistic effect from the Ni–Fe redox coupling within nanomaterials for the $CO_2RR$. However, the Ni-sulfide-based nanomaterials appear less stable than other TMS in the $CO_2RR$ and thus show lower efficiencies.

### 5.3. $FeS_x$

Zakaria et al. studied the use of FeS surfaces for the electrochemical hydrogen evolution and $CO_2$ reduction reactions [181]. The authors investigated a greigite ($Fe_3S_4$)-modified electrode, which was oxidized in an HER reaction to form a 60/40% mixture of greigite and goethite (FeOOH). The results in that study showed that the HER reaction was slowed dramatically in saturated $CO_2$ solution under reducing conditions. In addition, the binding indicated that the $HCO_3^-$ was surface-bound to the FeS [181]. Roldan et al. investigated the application of bio-inspired $CO_2$ conversion using FeS catalysts [182]. The study was focused on the application of $Fe_3S_4$ (greigite). In that work, the reduction of $CO_2$ was performed in a saturated solution at pHs of 4.5, 6.5, and 10.5, which showed that the formic acid was the major product at all pHs studied, with methanol, acetic acid, and pyruvic acid as the minor products. The authors used DFT calculations to study the $CO_2$ reduction, which indicated that the predominance of the formation of formic acid was due to the solution species of either $CO_2$ (neutral molecule), $HCO_3^-$ at intermediate pHs, or $CO_3^{2-}$ at high pHs. The molecule interacted with either the 001 or the 111 surface of the $Fe_3S_4$ particles [182]. Vladimirov et al. investigated the reduction of $CO_2$ on pyrite as a mechanism of abiogenic formation of organic molecules [183]. The authors used pyrite as a rotating disc electrode to form formate at high pressure (50 atm). As the potential applied to the electrode increased, the concentration of formate formed in solution increased, while a maximum current efficiency of 0.12% was observed at $-1$ V. The authors proposed that the reaction occurred at deep ocean vents (black smokers) on the ocean floors.

Simon et al. developed a fast microwave process to synthesize Ni-substituted $Fe_3S_4$ nanosheets for $CO_2$ reduction to CO [184]. The authors showed a faradaic efficiency for the $NiFe_3S_4$ nanomaterial of 6% at a potential of $-0.7$ V. However, at lower potentials, below $-0.8$ V, the HER reaction became predominate. Pellumbi et al. studied the effect of S/Se exchange on Fe/Ni pendalite for the electrocatalytic reduction of $CO_2$ to form CO [185]. The results in that study showed that the introduction of higher amounts of Se increased the formation of CO. Using galvanostatic conditions in acetonitrile, $Fe_{4.5}Ni_{4.5}S_4Se_4$ faradic efficiencies of approximately 100% were achieved at current densities of 50 mA/cm$^2$ and 100 mA/cm$^2$. Senthamaraikannan and Lim investigated the reduction of $CO_2$ to $C_1$ and $C_2$ products over sulfur deficient mackinawite (FeS) using DFT calculations [186]. The results showed that the adsorption energy for $CO_2$ onto pristine FeS was $-0.17$ eV, and on sulfur deficient surfaces $-1.62$ eV, which showed preferential binding and reduction of S-deficient surfaces. The favored product was methane in potential-limiting $^*HCO \rightarrow CH_2O$ ($\Delta G = 1.27$ eV) and $OHCH_2O^* \rightarrow CH_2O$ ($\Delta G = 0.78$ eV) reverse water–gas shift and formate pathways, respectively. For the formation of the $C_2$ species, a high concentration of $CO^*$ on the surface was required to yield ethane.

The preliminary studies reported on the use $Fe_yS_x$ nanomaterials in $CO_2RR$ have been used in an attempt to explain the origin of early life on earth. However, the application of $Fe_yS_x$ nanomaterials in electrocatalysis has shown great promise in developing value-added products from the $CO_2RR$. $Fe_yS_X$ and corresponding composite nanomaterials have been shown to produce large organic molecules, up to pyruvic acid ($C_3$ compound), not observed with any other TMS. However, formate/formic acid was the major product observed using Fe-based nanomaterials in $CO_2RR$.

### 5.4. CoS

Although CoS is a well-known catalyst for hydrogenation reaction, there was very limited information in the literature for the use of CoS in the electrocatalytic reduction of

$CO_2$. Yan et al. investigated $Co_3S_4$–$Co_3O_4$ core–shell octahedron nanomaterials for carbon reduction reactions as well as other applications [187]. The composite nanomaterial showed a high faradaic efficiency of 85.3% for the generation of formate at a potential of 0.64 V. The $Co_3S_4$ nano-needles showed a 60% faradic efficiency at $-0.6$ while $Co_3O_4$ showed approximately 35% faradaic efficiency at $-0.7$ V. Compared to other materials, the Co–S showed a relatively low faradaic efficiency for the $CO_2RR$.

As can be illustrated in Table 2 first-row transition metal sulfide nanomaterials and their associated nanocomposites have shown great diversity in the application to the $CO_2RR$. All the common products observed in the $CO_2RR$ are observed in the $CO_2RR$ using the first-row transition elements, which include CO, $CH_3OH$, $CH_4$, CHCOO/CHCOO-, $CH_3COOH$, $C_2H_4$, $CH_3CH_2OH$, and a higher carbon compound $CH_3COCOOH$. The results showed the diversity of the catalytic capability of the first-row transition elements in the $CO_2RR$.

**Table 2.** Summary of products in the $CO_2RR$ using the various TMS from the first-row transition element nanomaterials and respective associated nanocomposites.

| $CO_2RR$ Product | Material | Reference |
|:---:|:---:|:---:|
| CHCOO$^-$ | $CuS_X$ | [168] |
| | CuS | [169] |
| CHOOH | CuS | [170] |
| CHOOH | $Cu_2S$/CuS | [171] |
| | CuS–$C_3N_4$ | [172] |
| | $Cu2_S$ | [173] |
| | $CuS_X$ | [174] |
| $CH_4$ | CuS-nanosheet | [175] |
| CO, $CH_4$, $C_2H_4$, CHOO$^-$ | CuS | [176] |
| CO | NiS | [177] |
| $CH_3OH$ | $FeS_2$/NiS | [178] |
| $CH_4$ | NiS/$FeS_2$ | [179] |
| CO | NiS–C | [180] |
| $CH_3OH$, $CH_3CHOOH$, $CH_3COCOOH$ | $Fe_3S_4$ | [182] |
| CHOO$^-$ | $FeS_2$ | [183] |
| CO | $Fe_3S_4$ | [184] |
| CO | $Fe_{4.5}Ni_{4.5}S_4Se_4$ | [185] |
| CHCOO$^-$ | $Co_3S_4$–$Co_3O_4$ | [187] |

## 6. Non-Layered Later Transition/Metal Sulfide Nanomaterials and Their Associated Nanocomposites Applied to $CO_2$ Reduction Reactions

### 6.1. $Bi_xS_y$- and Bi-Based

Su et al. investigated the reduction of $CO_2$ over $Bi_2S_3$ nanosheets of carbon paper [188]. The authors synthesized nanosheets with an average size of 10 nm for the reduction of $CO_2$ in aqueous solutions. The results indicated that the nanosheets had a better performance than the bulk material, as well as showed high selectivity towards the formation of formate and high stability over 5 h of electrolysis. Hu et al. investigated a layered $Bi_2O_3$/$Bi_2S_3$ with a thickness of approximately 10 nm for the reduction of $CO_2$ to form formate [189]. The composite material had a faradaic efficiency of 85.5% for formate production with a current density of 14.1 mA/$cm^2$ at a voltage of $-1.0$ V. In that work, the experiments were performed using FTIR and showed that the rate-determining step in the formation of formate from $CO_2$ was the formation of the $HCOO^*$ intermediate. Yang et al. studied the sulfurization of bismuth oxide ($Bi_2O_3$) to generate a $Bi_2S_3$–$Bi_2O_3$ hybrid material for the generation of formic acid from $CO_2$ [190]. The hybrid material was synthesized using a partial precipitation method to generate interfaces between the $Bi_2S_3$ and $Bi_2O_3$ nanomaterials. The results showed that the production of formic acid was preferential over the formation of CO. The nanomaterial showed a faradaic efficiency of greater than 90% with an over-potential of 0.7 V with a current density of 6.14 mA/$cm^2$. Furthermore, the catalyst showed excellent stability/durability over 24 h of reaction. The $BiS_x$ nanomaterials and $BiS_x$/$Bi_2O_3$ composite have been shown to be effective at multiple $CO_2$ reduction reactions.

However, the most common reaction observed was the formation of formate/formic acid. The composite/hybrid oxide/sulfide nanomaterials also appear to favor the formation of formate/formic acid.

Zhan et al. investigated the use of $Bi_2S_3$ as an electrochemical reduction catalyst for the formation of formate from $CO_2$ [191]. The authors synthesized $Bi_2S_3$ nanorods which achieved a faradic efficiency of greater than 90% for the formation of formate. The authors used a potential range of $-0.9$ to $-1.2$ V while the catalyst showed prolonged stability for 12–20 h with a current density of 40 mA/cm$^2$. Ren et al. used a $Bi_2S_3$/ZiIF-8 (a zeolite) for the electrocatalytic reduction of $CO_2$ to form formate [192]. The results showed a faradaic efficiency of 74.2% at an over-potential of $-0.7$ V (compared to the reversible hydrogen electrode (RHE)) and current density of 16.1 mA/cm$^2$. The experiments were performed using UV diffuse reflectance spectroscopy, which indicated that the high reactivity and selectivity were a photo-electrochemical reduction.

Shao and Lui prepared a $Bi$/$Bi_2S_3$ catalysts for the reduction of $CO_2$ to form CO, formate, and performed the HER [193]. The generation of formate showed a maximum faradaic efficiency of 85% at an over-potential of $-1.0$ V (RHE) with current density of 17 mA/cm$^2$. The catalyst showed a good stability over 12 h of reaction. The authors concluded that the presence of $Bi(0)$ in the sample increased the conductivity of $Bi_2S_3$ while the selectivity towards formate was due to the sulfur and the synergistic effect between $Bi_2S_3$ and the $Bi(0)$.

### *6.2. CdS*

Li et al. investigated $CO_2$ reduction on nanorod CdS particles [194]. The study showed that the CdS nanorods were stable in the production of CO for over 40 h of reaction with a faradaic efficiency of 95% in aqueous solutions. The authors noted that a current density of 10 mA/cm$^2$ at an over-potential of $-0.55$ V was effective for $CO_2$ reduction. Additionally, a high selectivity of the CdS nanorods was reported, which was due to the 0002 face with sulfur vacancies [194]. The doping of CdS nanorods with Ag+ ions was investigated by Dong et al. for the electrochemical reduction of $CO_2$ to CO [195]. The results indicated that the doping of the CdS nanorods with the $Ag^+$ ions created S vacancies in the nanorods. The CdS–$Ag^+$ nanorods showed a faradaic efficiency of approximately 87% and a current density of 53.7 mA/cm$^2$. The Ag-doped CdS nanorods showed approximately double the faradaic efficiency of the CdS nanorods alone and approximately 1.5 times the faradic efficiency of CdS nanorods with S vacancies. Although CdS is a catalytic and photoactive material, there are insufficient data reported on the use of CdS in $CO_2$RR. Perhaps this is due to the toxicity of the Cd ion.

### *6.3. SnS$_x$*

Bai et al. studied SnS$_x$ supported on carbon cloth for the electrocatalytic reduction of $CO_2$ [196]. The authors used chalcogel as a precursor for the synthesis of SnS$_{0.09}$ and SnS$_{0.55}$ supported on carbon cloth. The SnS$_{0.55}$ showed a 93% faradaic efficiency to produce formate at a potential of $-1.1$ V, with a current density of 28.4 mA/cm$^2$. Li et al. studied the formation of formate from $CO_2$ using Sn/S derived from SnS$_2$ nanosheets [197]. The authors synthesized SnS$_2$ nanosheets supported on graphene oxide, which were capable of forming formate at over-potentials of 0.23 V. However, the optimum formation of formate occurred at a current density of 13.5 mA/cm$^2$, faradaic efficiency of 85% and over-potential of 0.68 V. The examination of the catalysis showed the presence of reduced Sn in the catalyst, which was formed from the cathodic reduction, but the enhanced performance was attributed to the presence of SnS$_2$. Liu et al. investigated methods to reduce sulfur dissolution and enhance the electrocatalytic reduction of $CO_2$ to formate on SnS [198]. The authors investigated In doping of the SnS structure which reduced the dissolution of the SnS. The In–SnS on carbon catalysis showed a faradaic efficiency of 97% for formate production at an over-potential of $-0.6$ V and a current density of 37 mV/cm$^2$.

Zhang et al. investigated InO–$SnS_2$ nanosheets for the electrochemical reduction of $CO_2$ to formate [199]. The authors performed the synthesis using an $InSn(OH)_6$ precursor which was functionalized into the In–O–$SnS_2$ nanosheets. The catalysts showed a faradaic efficiency to produce formate of 89% at a current density of 22.7 mA/$cm^2$ and an over-potential of 1.0 V. The unmodified $SnS_2$ nanosheets showed a current density of 9.4 mA/$cm^2$ and a faradaic efficiency for formate production of 18%. The DFT calculations indicated a synergistic effect of the In–O interacting with the $SnS_2$ creating oxidized Sn sites next to In sites, which reduced the activation energies and speeded up the reaction [199].

Li et al. prepared a Cu–SnS catalyst for the formation of formate from $CO_2$ [200]. The authors made a precatalyst consisting of a $Cu_2SnS_3$. The as-prepared catalyst showed a faradaic efficiency of 96% at a current density of −241 mA/$cm^2$. The authors also performed DFT calculations which showed preferred formation of formate by the $HCOO^*$ intermediate on the $CuSnS_x$ catalyst. The pristine $SnS_x$ and $SnS_x$ composite materials showed a preference for the formation of formate in $CO_2RR$. This indicates that these materials have a preference to form the $HCOO^*$ radical species over the other radical species. The studies indicate that SnS is not a great material for use in the $CO_2RR$; however, doping SnS with a secondary compound or ion can increase the faradaic efficiency tremendously, which shows great promise in controlled $CO_2RR$.

### 6.4. AgS

Lui et al. investigated the reduction of $CO_2$ in an ionic liquid using AgS [201]. The study utilized a cell with a Nafion membrane with aqueous solution voltages ranging from −0.144 to approximately −1.156 V (RHE), and in the ionic liquid the potential ranged from −0.136 to −1.164 V with a scan rate of 50 mV $s^{-1}$. The authors noted that the formation of CO, $H_2$, and $CH_4$ were the only reactions with a combined faradaic efficiency of 100%. However, the CO formation had a maximum efficiency of 92% around −0.864 V. In the presence of $KHCO_3$ the efficiency was lower. Ye et al. investigated the interaction of $CO_2$ with $Ag_2S$/Ag for the electrocatalytic formation of CO [202]. The study showed that the hybrid $Ag_2S$/Ag heterogenous catalyst achieved a large current density of approximately 422 mA/$cm^2$ at an over-potential of −0.70 V. However, a steady current density of 244 mA/$cm^2$ was achieved at an over-potential of −0.49 V, which gave a 99% faradaic efficiency. Chen et al. on the other hand investigated $Ag_2S$ on Au as an heterogenous catalyst for $CO_2$ reduction to CO [203]. The authors observed a faradaic efficiency of 94.5% at an over-potential of −0.8 V. In addition, the catalyst showed a current density of 9.2 mA/$cm^2$ with stability over 30 h. The authors performed DFT calculations, which indicated that the surface was favorable to the formation of $COOH^*$ radical species.

Shen et al. generated $Ag_2S$ electrocatalysts for the reduction of $CO_2$ in organic media [204]. The authors synthesized the $Ag_2S$ through the electro-oxidation of Ag in an aqueous solution on a Ag foil. In that work, the $Ag_2S$/Ag catalyst was considered as an effective electrocatalyst in propylene carbonate/tetrabutylammonium perchlorate [204]. The reaction showed a faradaic efficiency of 92% over 4 h of reaction at a current density of 9.85 mA/$cm^2$ and an over-potential of −2.3 V (compared to Fe/Fe+). Yang et al. investigated the in situ growth of Ag/$AgS_2$ nanowire clusters in a $H_2S$ plasma for the formation of CO from $CO_2$ [205]. The authors used a $H_2S$ plasma to treat a Ag foil, which generated Ag/$Ag_2S$ nanowire clusters. The nanocluster wires showed faradaic efficiencies of 83% at an over-potential of −0.4 V and a current density of 3 mA/$cm^2$. The highest faradaic efficiencies were observed with larger nanowires with diameters of 100 nm and lengths of microns. Zeng et al. investigated $Ag_2S$ on reduced graphene oxides for the electrocatalytic reduction of $CO_2$ to CO [206]. The authors used a hydrothermal method to produce the $Ag_2S$ on S/N-doped rGO. The $Ag_2S$ on S/N-doped rGO showed a selectivity of 87.4% towards producing CO from $CO_2$. In addition, the catalyst showed stability over 40 h at a current density of 70 uA/$cm^2$.

Cheng et al. investigated the electroreduction of $CO_2$ using $Ag_2S$ on CdS nanorods [207]. The authors studied the formation of CO, which was very low on CdS. However, the appli-

cation of $Ag_2S$ nanodots promoted the formation of CO at 5 wt% loading at a current density of 10.6 mA/cm$^2$ where 95% faradaic efficiency was achieved, which was approximately 93% enhancement over the pristine CdS.

The use of silver sulfide as a $CO_2RR$ catalyst has typically shown the production of CO in the reaction. The generation of CO seems independent of the doping material or the support used for the $Ag_2S$. This is a very interesting result as it shows controlled electrocatalysis in the $CO_2RR$ reaction towards a single product.

### 6.5. In Sulfides

Cai et al. designed ultrathin $ZnIn_2S_4$ nanomaterials grown on N-doped carbon cloth for the electrocatalytic reduction of $CO_2$ and the formation of ethanol [208]. The catalyst showed a 42% faradaic efficiency towards the formation of ethanol at a potential of $-0.7$ V in a $CO_2$-saturated solution of 0.5 M $KHCO_3$. The authors also performed DFT calculations which showed that the N-doped carbon cloth promoted the CO–CO coupling process leading to the formation of ethanol. Chi et al. investigated the stabilization of indium sulfide with Zn for the electrocatalytic reduction of $CO_2$ to from formate [209]. The $ZnIn_2S_4$ nanomaterial could produce formate with a 99% faradaic efficiency. The catalyst operated at a current density of 300 mA/cm$^2$ for over 60 h without decay. In addition, the authors performed computational studies which showed that the Zn increased the covalency of the In–S bonds, which formed a catalytic site to activate both $H_2O$ and $CO_2$ and gave the HCOO$^*$ intermediate, and thus formate was the product of the reaction.

There are only a few studies reported on the use of InS as an electrocatalyst in $CO_2RR$, which have shown a variety of products such as formate/formic acid and ethanol. Zn doping appears to push the reaction towards formate, as well as a being a high-stability catalytic material. In fact, supporting the $ZnIn_2S_4$ on carbon can push the reaction towards ethanol production, which may indicate that there is a contribution in the reaction from the carbon support.

The sulfides of the later transition elements as well as the metal (non-transition) elements associated with rows 2 and 3 of the periodic table show a promise for application in the $CO_2RR$. As can be seen in Table 3, the majority of the common products observed in the $CO_2RR$ are the sulfides and their composites, which include CO, $CH_4$, $CHCOO/CHCOO^-$, $C_2H_4$, and $CH_3CH_2OH$. The results showed the catalytic capability and the ability to specifically react with the C/O to break C–O bonds and form C–H bond to give specific products.

**Table 3.** Summary of products in the $CO_2RR$ using various sulfides from the later transition/metal elements nanomaterials and their associated composite nanomaterials.

| $CO_2RR$ Product | Material | Reference |
|---|---|---|
| $CHOO^-$/$CHOOH$ | $Bi_2S_3$ | [188] |
| | $Bi_2O_3$/$Bi_2S_3$ | [189] |
| | $Bi_2S_3$–$Bi_2O_3$ | [190] |
| | $Bi_2S_3$-nanorods | [191] |
| | $Bi_2S_3$–ZiIF-8 | [192] |
| | $Bi$/$Bi_2S_3$ | [193] |
| CO | CdS | [194] |
| | CdS–Ag | [195] |
| $CHOO^-$ | $SnS_x$ | [196] |
| | $SnS_2$ | [197] |
| | SnS | [198] |
| | $InO$–$SnS_2$ | [199] |
| | $Cu$–$SnS$ | [200] |
| CO, $CH_4$ | AgS | [201] |
| | $Ag_2S$/Ag | [202] |
| | $Ag_2S$/Au | [203] |
| | $Ag$/$AgS_2$ | [205] |
| | $Ag_2S$ | [206] |
| | $Ag_2S$/CdS | [207] |
| $CH_3CH_2OH$ | $ZnIn_2S_4$–$C_3N_4$ | [208] |
| $CHOO^-$ | $ZnIn_2S_4$ | [209] |

## 7. Conclusions/Future Perspectives

The TMS are very capable of reducing $CO_2$ to form value-added products such as CO, $HCOO^-$/HCOOH, $CH_3OH$, $CH_3CH_2OH$, $CH_4$, and $C_2H_4$. There are a few studies that showed even larger molecule formation. However, the TMS-composite materials showed much more promise in the development of value-added products from $CO_2$. For example, the doping of individual TMS with various materials showed much higher faradaic efficiencies than the pristine materials, such as was observed with $MoS_2$, CdS, $In_2S_4$, $Ag_2S$, SnS, CuS, and among the other sulfides presented in the review. The doping of materials changes the ability to attract $CO_2$ and the binding strength of the reduction products to the surface and allows for higher efficiencies.

The Future of the electrocatalytic reduction of $CO_2$ will be dependent on the development of nanomaterials that are not only TMS but composites of two or more systems. In addition, very favorable results have been observed especially in the substitution of Ni into $Fe_yS_x$ and Fe into the $Ni_yS_x$. The substitution appears to create a redox couple within the material and the $CO_2RR$ appears to be enhanced. The addition of carbon and N-doped carbon appears to have a large effect on the catalytic activity of the material. The synergistic effect of catalytic supports as well as the development of hybrid materials will be a major push forward in the use of TMS in $CO_2RR$. In addition, the movement of the reactions away from acidic pHs will increase the use of sulfides in electrocatalysis while the movement away from acidic pHs will improve the stability of the TMS in the reaction media. Furthermore, a movement away from aqueous media will also increase the stability of TMS and thus the longevity of both the reaction and the TMS' materials. By controlling the structure of the catalytic material and the support, very controlled catalytic reactions can be achieved. Through the development of new composite materials, catalytic reactivities will be increased, which was observed in the HDS reaction catalysis. The synthesis of Co/Ni on $MoS_2$ and $WS_2$ catalysts resulted in synergistic activities and higher reaction rates for the catalysts. As more composite materials are used, the field and application of TMS in the $CO_2RR$ will grow, while the application of TMS in electrocatalysis is in its infancy as can be observed by the lower number of papers reported on using TMS in $CO_2RR$ compared to other types of materials.

The most effective catalysts will be those which can do both the $CO_2RR$ but also perform the HER reaction simultaneously. As it was discussed in this review, the hydrogenation of the reduced $CO_2$ species is a key parameter in controlling the reaction pathways, and ultimately the final reaction products.

**Author Contributions:** Conceptualization, J.P.; writing—original draft preparation, J.P.; writing—review and editing, J.P. and M.A.; funding acquisition, J.P. and M.A.; All authors have read and agreed to the published version of the manuscript.

**Funding:** J.G. Parsons acknowledges and is grateful for the support provided by a funding from the UTRGV Chemistry Departmental Welch Foundation Grant (Grant No. BX-0048), and M. Alcoutlabi acknowledges funding from NSF PREM (DMR-2122178) Partnership for Fostering Innovation by Bridging Excellence in Research and Student Success.

**Institutional Review Board Statement:** Not applicable.

**Informed Consent Statement:** Not applicable.

**Data Availability Statement:** Not applicable.

**Conflicts of Interest:** The authors declare no conflict of interest.

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
