# Peer review of "The Application of Transition Metal Sulfide Nanomaterials and Their Composite Nanomaterials in the Electrocatalytic Reduction of CO2: A Review"

_applsci, doi:10.3390/app13053023_

Round 1

Reviewer 2 Report

Reviewer’s Comments:

The manuscript “Application of Transition Metal Sulfides and their composite Nanomaterials in the electrocatalytic reduction of CO2.” is a very interesting work. This paper presents the Electrocatalysis has become an important topic in various areas of research, including chemical catalysis, environmental research, and chemical engineering. There have been a multitude of different catalysts used in electrocatalytic reduction of CO2 (CO2RR) which include large classes of materials such as transition metal oxide nanoparticles (TMO), transition metal nanoparticles (TMM), carbon-based nanomaterials, and transition metal sulfides (TMS), as well as porphyrins and phthalocyanine molecules. This review is focused on the CO2RR and the main products produced using transition metal sulfide nanomaterials. The main reaction products of the CO2RR include carbon monoxide (CO), formate/ formic acid (HCOO-/HCOOH), methanol (CH3OH), ethanol (CH3CH2OH), methane (CH4), and ethene (C2H4). While I believe this topic is of great interest to our readers, I think it needs major revision before it is ready for publication. So, I recommend this manuscript for publication with major revisions.

1. In this manuscript, the authors did not explain the importance of the composite Nanomaterials in the introduction part. The authors should explain the importance of composite Nanomaterials.

2) Title: The title of the manuscript is not impressive. It should be modified or rewritten it.

3) Correct the following statement “The products of the CO2RR have been linked the metal sulfide catalyst used, which controls the intermediated and the reaction pathway. Both experimental and computational methods have been utilized to determine the CO2 binding and chemically reduced intermediates, which drive the reaction pathways for the CO2RR.”.

4) Keywords: The composite Nanomaterials is missing in the keywords. So, modify the keywords.

5) Introduction part is not impressive. The references cited are very old. So, Improve it with some latest literature like 10.1016/j.apcatb.2020.119097, 10.1016/j.jece.2021.105534

6) The authors should explain the following statement with recent references, “However, the bulk crystals showed degradation by loss of S as H2S and was not noted in the thin film studies”.

7) Add space between magnitude and unit. For example, in synthesis “21.96g” should be 21.96 g. Make the corrections throughout the manuscript regarding values and units.

8) The author should provide reason about this statement “The results in that study showed that high number of exposed edge sites on the MoS2 provided a high number of active sites and lowered the onset potential with a 93% Faradaic efficiency with an over potential of 0.59 V”.

9. Comparison of the present results with other similar findings in the literature should be discussed in more detail. This is necessary in order to place this work together with other work in the field and to give more credibility to the present results.

10) Conclusion part is very long. Make it brief and improve by adding the results of your studies.

11) There are many grammatic mistakes. Improve the English grammar of the manuscript.

Reviewer 3 Report

In the following submission, “Application of Transition Metal Sulfides and their composite  Nanomaterials in the electrocatalytic reduction of CO2, the authors claimed to have reviewed the recent progress on electrocatalytic reduction of CO2 for the formation of value-added products using transition metal sulfide nanomaterials. Although a large number of review articles have been published recently on metal sulfides including their layered counterparts that emphasize the challenges and their applications in various fields but more advanced level knowledge can facilitate the readers with existing challenges and their solutions as well as how transitional metal sulfides-based hybrid materials can be modified at molecular level to achieve better performance for different applications. The present review article mainly summarizes the recent work on the electrocatalytic application of TMS but failed to describe the general synthesis, characterization and challenges associated with these materials systematically, thus in my opinion can only be accepted for publication in “Applied Sciences” if it can be revised (Major revision) carefully considering the current requirement of review articles. Additional questions are listed below.

1.        Abstract should briefly describe the contents of the review; such this review contains ….

2.        Some paragraphs are very long and arranged haphazardly which should be organized properly.

3.        The introduction is very weak should be strengthen by adding importance and specific features of TMS and their composites in electrocatalytic applications.

4.        Include few more sections to describe few common preparation and characterization methods of TMS

5.        I believe figures are less, thus few figures from the cited literature would enhance the quality of review

6.        Instead of putting TMS in different sections (section 4 to 12), all these TMS should be included under a common heading such as TYPES OF TMS FOR ELECTROCATALTYC APPLICATIONS etc., and represent with different subheading

7.        The authors should also share the knowledge about most preferred TMS in electrocatalytic application and should give the reasons.

8.        line 184 reference is wrong

9.        There are few grammatical and spelling mistakes in the manuscripts, revise the manuscript carefully.

10.     Conclusion should be improved by including future prospects and challenges of TMS in electrocatalysis

      I.          The authors are suggested cite these articles in this review:

a.      Khan, Mujeeb, et al. "Graphene based metal and metal oxide nanocomposites: synthesis, properties and their applications." Journal of Materials Chemistry A 3.37 (2015): 18753-18808.

b.     Ashraf, Muhammad, et al. "A High‐Performance Asymmetric Supercapacitor Based on Tungsten Oxide Nanoplates and Highly Reduced Graphene Oxide Electrodes." Chemistry–A European Journal 27.23 (2021): 6973-6984.

c.      Gherab, K., et al. "Fabrication and characterizations of Al nanoparticles doped ZnO nanostructures-based integrated electrochemical biosensor." Journal of Materials Research and Technology 9.1 (2020): 857-867.

Round 2

Reviewer 1 Report

 Check the following before publish-

1. The formation of CH4 typically occurs through the formation of the CO* radial, which is hydrogenated to form the HCO* radical species followed by subsequent hydrogen addition reactions to give CH4 [66]. 

2. However, most of the work that has been performed in the filed does not focus on the specific application of TMS and their composite nanomaterials in the CO2RR. 

Author Response

The authors would like to thank the reviewers for all their feedback and valuable comments on the manuscript. Based on reviewer’ suggestions and comments, we have revised the manuscript and made the appropriate changes accordingly. 

Comment:  The formation of CH4 typically occurs through the formation of the CO* radial, which is hydrogenated to form the HCO* radical species followed by subsequent hydrogen addition reactions to give CH4 [66].

Response: We have corrected the error in the sentence which now reads as follows: “ The formation of CH4 typically occurs through the formation of the CO* radical, which is hydrogenated to form the HCO* radical species followed by subsequent hydrogen addition reactions to give CH4 [66].”

Comment: However, most of the work that has been performed in the filed does not focus on the specific application of TMS and their composite nanomaterials in the CO2RR.

Response: we have corrected the line of the manuscript it now reads: “However, most of the work that has been performed in the field does not focus on the specific application of TMS and their composite nanomaterials in the CO2RR.”

Reviewer 3 Report

The authors have addressed all the comments raised by the reviewer, now it can be published in the journal

Author Response

The authors would like to thank the reviewers for all their feedback and valuable comments on the manuscript. Based on reviewer’ suggestions and comments, we have revised the manuscript and made the appropriate changes. 

Comment: The authors have addressed all the comments raised by the reviewer, now it can be published in the journal

Response: We thank the reviewer for his comments and time.